# Do Not Blindly Imitate the Teacher: Loss Perturbation for Knowledge Distillation

## Abstract

Knowledge distillation (KD) is a popular model compression technique to transfer knowledge from large teacher models to a small student model. Typically, the student learns to imitate the teacher by minimizing the KL divergence of its output distribution with the teacher's output distribution. We argue that such a learning objective is sub-optimal because there exists a discrepancy between the teacher's output distribution and the ground truth label distribution, and forcing the student to blindly imitate the unreliable teacher output distribution leads to inferior performance. To this end, we propose a novel knowledge distillation objective *PTLoss* by first representing the vanilla KL-based distillation loss function via a Maclaurin series and then *perturbing* the leading-order terms in this series. This perturbed loss improves the student generalizability by effectively distilling knowledge from a shifted distribution closer to the ground truth data. We also propose a method to compute this shifted teacher distribution, named *Proxy Teacher*, which enables us to select the perturbation coefficients in PTLoss. We theoretically show the perturbed loss reduces the deviation from the true population risk compared to the vanilla KL-based distillation loss functions. Experiments on three tasks with teachers of different scales show that our method significantly outperforms vanilla distillation loss functions and other perturbation methods.

## 1 Introduction

Deep neural networks (DNNs) have achieved enormous success due to their massive sizes, expressive power, and the availability of large-scale data for model training. Accompanied by their success is the increasing need of deploying such large-scale DNNs on resource-limited devices. Knowledge distillation (KD) is a widely-used model compression technique, which distills knowledge from large teacher models into a much smaller student model to preserve the predictive power of teacher models (Buciluǎ et al., 2006; Hinton et al., 2015). In the teacher-student learning paradigm of KD, the student model is encouraged to imitate the teacher models' outputs on a distillation dataset.

The typical training objective in KD such as KL loss (Hinton et al., 2015; Menon et al., 2021; Stanton et al., 2021) encourages the student's outputs to be close to the teacher's outputs as much as possible. This implicitly assumes the teacher's outputs on the distillation data are perfect. However, the teacher's predictive distributions can be biased from the ground truth due to various factors, such as the inductive bias encoded in the teacher's model architecture, miscalibration in the training procedure (Menon et al., 2021), or the bias in the source dataset used for learning the teacher model (Liu et al., 2021; Lukasik et al., 2021). Enforcing the student to blindly imitate the teacher's outputs can make the student inherit such biases and produce suboptimal predictions.

To overcome this challenge, one commonly used approach is to scale the teacher's logits via a temperature parameter (Hinton et al., 2015). Menon et al. (2021) show that a proper temperature can improve the quality of the teacher model's predictive distribution, but the shifting space offered by the temperature scaling is limited, and the optimal temperature value relies on expensive grid search. Along a separate line, label smoothing (Szegedy et al., 2016) is proposed as a general technique to regularize the neural networks, and modulated loss functions (Lin et al., 2017; Leng et al., 2022) are designed to address several statistical issues (*e.g.,* overfitting issues and data imbalance) in model training. However, there lack works that explore tailoring such techniques for more robust knowledge distillation.

We propose *PTLoss* for knowledge distillation, which revises the vanilla loss function in knowledge distillation to implicitly create a debiased teacher distribution closer to the ground truth. Instead of forcing an out-and-out imitation of the original teacher model, we relax the KL loss and add perturbations to the distillation objective. Specifically, we approximate the standard KL loss using the Maclaurin series, which allows us to construct a more flexible objective and to perturb the leading-order terms. To determine the perturbation extent, we design a method to compute the equivalent distribution of the implicitly shifted teacher by perturbation, namely *Proxy Teacher*. With the computed proxy teacher distribution, we measure the empirical deviation between the perturbed teacher and the ground truth data. It leads to a systematic searching strategy for the perturbation coefficients, *i.e.,* the near-optimal perturbation coefficients should minimize the deviation between distilled risk and population risk on the validation set.

Theoretically, we justify the effectiveness of the PTLoss by proving that it can reduce the deviation from the distilled empirical risk compared to KL loss. We draw a connection between the PTLoss and other perturbation method (*i.e.,* label smoothing(Szegedy et al., 2016)). We illustrate that the PTLoss can debias the teacher to produce higher-fidelity outputs via a finer-grained perturbation, while subsuming existing perturbation techniques as special cases. Experiments on three datasets with different-sized teacher models demonstrate the empirical advantage of the PTLoss. Moreover, the Proxy Teacher method for perturbation coefficient search significantly outperforms the PTLoss with random searched coefficients, which shows the superiority of this systematic parameter search method.

In summary, our key contributions are:

- A perturbed loss function PTLoss, which formulates the vanilla knowledge distillation loss in the form of Maclaurin series and perturbs it to improve the fidelity of teacher models;
- A Proxy Teacher method to solve the implicitly shifted teacher and to determine the perturbation coefficients in PTLoss;
- Theoretical analysis proves that we can lower the distilled empirical risk bound with PTLoss and establishes the connection with other perturbation methods;
- Comprehensive experiments on three public datasets with different-sized teacher models demonstrating the advantage of the PTLoss and the Proxy Teacher method.

## 2 RELATED WORK

### 2.1 KNOWLEDGE DISTILLATION

Knowledge distillation is first proposed in (Buciluǎ et al., 2006) to compress the large model ensembles to smaller, faster models without a significant performance drop. This technique is generalized by (Hinton et al., 2015), where the temperature parameter is introduced to smooth the prediction and the student loss and distillation loss are integrated. With the prevalence of pre-trained language models (Devlin et al., 2018), it becomes more urgently needed to distill such large models to deploy on edge devices with limited resources. For example, DistillBERT (Sanh et al., 2019) uses the teacher's soft prediction probability to train the student model; TinyBERT (Jiao et al., 2019) aligns the student's layer outputs (including attention outputs and hidden states) with the teacher's; MobileBERT (Sun et al., 2020) also adopts a layer-wise training objective and equips a bottleneck structure to distill from BERT-Large.

### 2.2 DISTILLATION THEORY

In parallel with the empirical success of the application of knowledge distillation, many works are devoted to answering its mechanism. Hinton et al. (2015) propose the teacher's soft labels can provide "dark knowledge" via weights on the wrong labels. Menon et al. (2021) present a statistical perspective on distillation, they observed that a good teacher model should be Bayesian to lower the variance of the student objective via the teacher's prediction distribution. Stanton et al. (2021) show the discrepancy between the teacher and the student regarding their output distribution and identify the optimization difficulty in knowledge distillation. Ji & Zhu (2020); Zhou et al. (2021); Hsu et al. (2021) study distillation from several different aspects, but there remains a gap between the theoretical analysis and the better distillation techniques.

## 2.3 LOSS DESIGN

Our work is also related to loss function design and learning. Lin et al. (2017) propose to reshape the cross-entropy loss to focus on the hard examples, but it cannot be directly applied in our setting because the data imbalance issue targeted by their approach do not directly correlated with the biased teacher model in knowledge distillation.Leng et al. (2022) propose to expand cross-entropy loss and focal loss to a linear combination of polynomial functions and study the Poly-1 formulation on computer vision tasks. However, the motivation is not clear under the fully supervised settings and they skirt around the problem of hyper-parameter search in a high-dimension space when the order of polynomials is high. Notably, TaylorGLO (Gonzalez & Miikkulainen, 2021) utilizes Covariance Matrix Adaptation Evolution Strategy (CMA-ES) to optimize multivariate Taylor parameterization of a loss function and learning rate schedule during training. But they fail to provide a principled analysis regarding the performance gain after perturbation. Instead, we theoretically and empirically demonstrate the necessity of adding perturbation to the learning objective for knowledge distillation, where a high-fidelity teacher is required to provide quality supervision for the student training.

## 3 PRELIMINARIES

We study knowledge distillation for classification tasks. For brevity, we discuss binary classification in this paper, but the formulation can be extended to multi-class settings.

In binary classification, we are given a training sample set $\mathcal{S} = \{(x_n, \mathbf{y}_n)\}_{n=1}^N \sim \mathbb{P}^N$, for distribution $\mathbb{P}$ over instances $\mathcal{X}$ and labels $\mathcal{Y} = \{0, 1\}^2$. Our goal is to learn a logits predictor $\mathbf{f} : \mathcal{X} \to \mathbb{R}^2$ with minimal risk:

$$R(\mathbf{f}) = \mathbb{E}_{(x,\mathbf{y})\sim\mathbb{P}}[\ell(\mathbf{y}, \mathbf{f}(x))]. \tag{1}$$

Here, $\ell(\mathbf{y}, \mathbf{f}(x))$ is the loss of predicting $\mathbf{f}(x) \in \mathbb{R}^2$ when the true label is $\mathbf{y} \in \mathbb{R}^2$. The predicted probabilities are $\mathbf{p}(x) = (\frac{e^{f_1(x)}}{e^{f_1(x)}+e^{f_2(x)}}, \frac{e^{f_2(x)}}{e^{f_1(x)}+e^{f_2(x)}})$, which is invariant to a constant shift to both $f_1(x)$ and $f_2(x)$. Thus we can simplify the $\mathbf{f}(x)$ as $(0, f(x))$.

In this paper, we consider the KL divergence:

$$\ell(\mathbf{y}, \mathbf{f}(x)) = \mathbf{y}^T \log(\mathbf{y}) - \mathbf{y}^T \mathbf{f}(x) + \log Z(\mathbf{f}(x)), \tag{2}$$

where $Z$ is the partition function $Z(\mathbf{x}) = \mathbf{1}^T e^{\mathbf{x}}$ for $\mathbf{x} \in \mathbb{R}^2$. For finetuning from true labels, we approximate the risk $R(\mathbf{f})$ via the empirical risk

$$\hat{R}(\mathbf{f}) = \frac{1}{N} \sum_{n\in[N]} \mathbf{y}_n^T \boldsymbol{l}(\mathbf{f}(x_n)) \tag{3}$$

for one-hot encoding $\mathbf{y}_n \in \{0, 1\}^2$, and vector of losses for each possible label $\boldsymbol{l}(\mathbf{f}(x)) \in \mathbb{R}^2$, which is defined as

$$\boldsymbol{l}(\mathbf{f}(x)) = [-\mathbf{f}(x) + \log Z(\mathbf{f}(x)) \cdot \mathbf{1}]. \tag{4}$$

The term $\mathbf{y}^T \log(\mathbf{y})$ is omitted in the above equation because it is always zero for one-hot hard labels.

For distillation from teacher's predictions, we first compute teacher class-probability estimates $\mathbf{p}^t(x) = (p^t(0|x), p^t(1|x))$, where $p^t(y|x)$ estimates how likely $x$ is to be classified as $y$. Then we train a student model to minimize the distilled risk

$$\tilde{R}(\mathbf{p}^t, \mathbf{f}; S) = \frac{1}{N} \sum_{n\in[N]} \left( \mathbf{p}^t(x_n)^T \boldsymbol{l}(\mathbf{f}(x_n)) + \mathbf{p}^t(x_n)^T \log(\mathbf{p}^t(x_n)) \right). \tag{5}$$

Note that each loss from the above risk is the widely used distillation loss, proposed in (Hinton et al., 2015). It is the KL divergence of the student output distribution from the teacher output distribution[1]:

$$\ell(\mathbf{p}^t(x_n), \mathbf{f}^s(x_n)) = \ell_{KL}\left(\mathbf{p}^t(x_n), \mathbf{p}^s(x_n)\right) = \mathrm{KL}\left(\mathbf{p}^t(x_n)\|\mathbf{p}^s(x_n)\right) = \sum_{j\in\{0,1\}} \mathbf{p}_j^t(x_n) \log\left(\frac{\mathbf{p}_j^t(x_n)}{\mathbf{p}_j^s(x_n)}\right), \tag{6}$$

---

[1]For simplicity, we assume the teacher model here does not contain temperature scaling, we will discuss the impact of different temperatures in the experiment section.

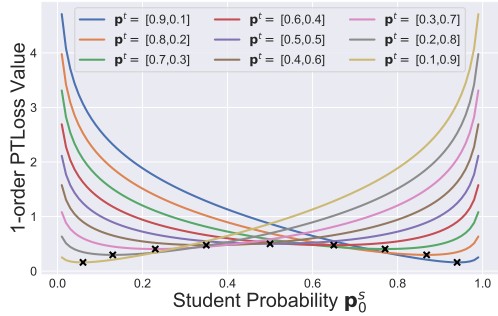 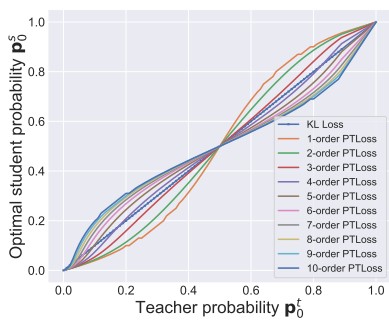

(a) The 1-order PTLoss ($M = 1, \epsilon_1 = 1, \gamma_1 = 1$) with different teacher probabilities $\mathbf{p}^t$. For each curve, $\mathbf{p}^t$ is fixed and we search the optimal student probability $\mathbf{p}_0^s$ on class 0. Black crosses denote $\mathbf{p}_0^s$ achieving the lowest loss value given $\mathbf{p}^t$.

(b) Comparison between the standard KL-loss with PTLoss with perturbation order from 1 to 10. Each perturbation coefficient $\epsilon_j$ and $\gamma_j$ are set as 2 for this plot.

Figure 1: Intuitive understanding of PTLoss via basic forms.

where $\mathbf{p}_j^t(x_n)$ and $\mathbf{p}_j^s(x_n)$ denote the probability of the $i$-th example $x_n$ belonging to the $j$-th class according to the teacher (student) model.

Instead of directly learning a student model $\mathbf{p}^s$ from $\mathcal{S}$, knowledge distillation learns the student $\mathbf{p}^s$ from a trained teacher model $\mathbf{p}^t$ by replacing the ground truth one-hot label $y_n$ with the teacher output probabilistic label estimates $\mathbf{p}^t(x_n)$. Then, the student model is learned by minimizing the *empirical distillation risk* $\tilde{R}(\mathbf{p}^t; \mathbf{p}^s, \mathcal{S}) \doteq \frac{1}{N} \sum_{n=1}^{N} \ell_{KL}(\mathbf{p}^t(x_n), \mathbf{p}^s(x_n))$.

## 4 PERTURBED DISTILLATION LOSS

Using KL divergence as the distillation loss (in short "KL loss") essentially assumes the teacher model is perfect and thus forces the student model to mimic the teacher's output label distribution. In reality, the teacher model can produce a biased estimate of label distribution and lead to a sub-optimal student model, as demonstrated by both empirical results (Müller et al., 2019) and theoretical analysis (Menon et al., 2021).

In this work, we present a novel distillation loss that relaxes the standard KL loss form and accommodates the distribution gap between (possibly biased) teacher output distribution and underlying ground truth distribution. Inspired by the PolyLoss (Leng et al., 2022), we propose to first replace the logarithmic terms in the standard KL loss with their corresponding Maclaurin series:

$$\log(1 - x) = -\sum_{m=1}^{\infty} \frac{x^m}{m}, \quad \log(x) = \log(1 - (1 - x)) = -\sum_{m=1}^{\infty} \frac{(1 - x)^m}{m}. \tag{7}$$

Then we perturb the polynomial terms with two perturbation coefficients $\gamma_m$ and $\epsilon_m$:

$$\log(1 - x) \approx -\sum_{m=1}^{\infty} (\frac{1}{m} + \gamma_m)x^m, \quad \log(x) \approx -\sum_{m=1}^{\infty} (\frac{1}{m} + \epsilon_m)(1 - x)^m. \tag{8}$$

Here, we essentially replace the original coefficient $\frac{1}{m}$ of the $m$-th order polynomial term in the standard KL loss to $(\frac{1}{m} + \gamma_m)$ or $(\frac{1}{m} + \epsilon_m)$. When $\gamma_m, \epsilon_m \in [-\frac{1}{m}, \infty]$, we can further separate these perturbation coefficients with the original coefficient and obtain:

$$\log(1 - x) \approx \log(1 - x) - \sum_{m=1}^{\infty} (\gamma_m)x^m, \quad \log(x) \approx \log(x) - \sum_{m=1}^{\infty} (\epsilon_m)(1 - x)^m. \tag{9}$$

Now, by replacing the term $\log \mathbf{p}_0^s(x_n)$ and $\log \mathbf{p}_1^s(x_n) = \log(1 - \mathbf{p}_0^s(x_n))$ in the standard KL loss Eq. 6 with Eqs. 7 and 9, we have:

$$\ell_{KL}(\mathbf{p}^t(x_n), \mathbf{p}^s(x_n)) = -\mathbb{H}(\mathbf{p}^t(x_n)) + \mathbf{p}_0^t(x_n)[-\log \mathbf{p}_0^s(x_n)] + \mathbf{p}_1^t(x_n)[-\log \mathbf{p}_1^s(x_n)]$$

$$\approx -\mathbb{H}(\mathbf{p}^t(x_n)) + \mathbf{p}_0^t(x_n)\left[-\log \mathbf{p}_0^s(x_n) + \sum_{m=1}^{\infty} \epsilon_m(1 - \mathbf{p}_0^s(x_n))^m\right]$$

$$+ (1 - \mathbf{p}_0^t(x_n))\left[-\log(1 - \mathbf{p}_0^s(x_n)) + \sum_{m=1}^{\infty} \gamma_m(\mathbf{p}_0^s(x_n))^m\right], \tag{10}$$

where $\mathbb{H}\left(\mathbf{p}^t(x_n)\right) = -\mathbf{p}^t(x_n)^T \log \mathbf{p}^t(x_n)$ is the entropy of the teacher output distribution and thus is student-independent. By further separating out perturbation coefficients on the right hand side of the above Eq. 10, we have:

$$\ell_{KL}\left(\mathbf{p}^t(x_n), \mathbf{p}^s(x_n)\right) \approx -\mathbb{H}\left(\mathbf{p}^t(x_n)\right) + \mathbf{p}_0^t(x_n)\left[-\log \mathbf{p}_0^s(x_n)\right] + \left(1 - \mathbf{p}_0^t(x_n)\right)\left[-\log(1 - \mathbf{p}_0^s(x_n))\right]$$
$$+ \mathbf{p}_0^t(x_n)\left[\sum_{m=1}^{\infty} \epsilon_m(1 - \mathbf{p}_0^s(x_n))^m\right] + (1 - \mathbf{p}_0^t(x_n))\left[\sum_{m=1}^{\infty} \gamma_m(\mathbf{p}_0^s(x_n))^m\right], \tag{11}$$

where the first line on the right hand side is essentially the original KL loss. Therefore, we finally define our perturbed distillation loss as:

$$\ell_{PT} \doteq \ell_{KL} + \mathbf{p}_0^t(x_n)\sum_{m=1}^{\infty} \epsilon_m\left(1 - \mathbf{p}_0^s(x_n)\right)^m + \left(1 - \mathbf{p}_0^t(x_n)\right)\sum_{m=1}^{\infty} \gamma_m(\mathbf{p}_0^s(x_n))^m. \tag{12}$$

If we only perturb the leading $M$-th order terms, then we have:

$$\ell_{PT-M} \doteq \ell_{KL} + \mathbf{p}_0^t(x_n)\sum_{m=1}^{M} \epsilon_m\left(1 - \mathbf{p}_0^s(x_n)\right)^m + \left(1 - \mathbf{p}_0^t(x_n)\right)\sum_{m=1}^{M} \gamma_m(\mathbf{p}_0^s(x_n))^m. \tag{13}$$

In vector form, it is

$$\ell_{PT-M} \doteq \ell_{KL} + \mathbf{p}^t(x_n)^T \sum_{m=1}^{M} \eta_m \odot \left(\mathbf{1} - \mathbf{p}^s(x_n)\right)^m, \tag{14}$$

where $\odot$ is the element-wise product, and $\eta_m = (\epsilon_m, \gamma_m)$.

Figure 1 shows some basic forms of the PTLoss to give an intuitive understanding of the loss function family and the perturbation extent. In Figure 1a, we present the simplest 1-order PTLoss to show how the perturbation influences the student training. For example, when $\mathbf{p}^t = [0.1, 0.9]$, the optimal student probability on class 0 is between 0 to 1. In general, This specific PTLoss makes a polarized adjustment to the student prediction. In Figure 1b, we show PTLoss with different perturbation orders, *i.e.,* $M \in \{1, 2, \cdots, 10\}$. For KL loss, the identical curve implies it encourages a total imitation of the teacher's behavior. On the other hand, the PTLoss relaxes the objective by perturbations and thus prevents the student model from overfitting the biased teacher. The perturbation granularity can be adjusted by the perturbation order and coefficients. Note that we set all the perturbation coefficients as 2 for simplicity in Figure 1b, but in practice, we use a systematic approach to determine the perturbation coefficients, as described in the below Section 4.1.

## 4.1 PROXY TEACHER

In this section, we theoretically prove that the introduced PTLoss allows for reducing the deviation from the true population risk. We first show that PTLoss implicitly transforms the teacher's predictions, then we select the minimizer in terms of the perturbation coefficients in PTLoss to bound the distilled empirical risk with introduced perturbation. Finally, we prove that PTLoss helps us get a closer estimation of the predictor to the oracle signal function.

### 4.1.1 THEORY OF DISTILLATION

We start from the population risk $R(\mathbf{f})$ defined in Eq. 1 and write it in a Bayes classifier form as

$$R(\mathbf{f}) = \mathbb{E}_x[\mathbb{E}_{\mathbf{y}|x}[\ell(\mathbf{y}, \mathbf{f}(x))]] = \mathbb{E}_x[\mathbf{p}^*(x)^T \boldsymbol{l}(\mathbf{f}(x))], \tag{15}$$

where $\mathbf{p}^*(x) = \mathbb{P}(\mathbf{y}|x)$ with $\mathbf{y} \in \{0, 1\}^2$ is the Bayes class probability distribution in the label space, and $\ell(\mathbf{y}, \mathbf{f}(x))$ and $\boldsymbol{l}(\mathbf{f}(x))$ were defined in Eqs. 2 and 4, respectively. Again, the term $\mathbf{y}^T \log(\mathbf{y})$ is omitted in the above equation because it is always zero for one-hot hard labels.

When labeled examples are limited, people learn the student $\mathbf{f}^s(x)$ by minimizing the distilled risk $\tilde{R}(\mathbf{p}^t, \mathbf{f}; \mathcal{S})$ to achieve low population risk $R(\mathbf{f}^s)$. For any teacher model's predictions $\mathbf{p}^t$, we have the following proposition:

**Proposition 1.** *Given a bounded loss $l^2$, a teacher model $\mathbf{p}^t$ with its distilled empirical risk defined in Eq. 5, and any predictor $\mathbf{f} : \mathcal{X} \to \mathbb{R}^2$, we have:*

$$\mathbb{E}\left[(\tilde{R}(\mathbf{p}^t, \mathbf{f}; \mathcal{S}) - R(\mathbf{f}))^2\right] \leq \frac{2}{N} \cdot \mathbb{V}\left[\mathbf{p}^t(x)^T \boldsymbol{l}(\mathbf{f}(x))\right] +$$
$$\mathcal{O}\left(\left(\mathbb{E}_x[\|\mathbf{p}^t(x) - \mathbf{p}^*(x)\|_2]\right)^2 + \mathbb{E}_x\left[\left(\mathbf{p}^t(x)^T \log \mathbf{p}^t(x)\right)^2\right]\right), \quad (16)$$

*where $\mathbb{V}[\cdot]$ denotes the variance of a random variable.*

We defer the proof to appendix. The above proposition states that the deviation between distilled risk and population risk depends on three terms: (1) variance of KL loss for a random example, (2) MSE between teacher's predictions $\mathbf{p}^t$ and the true distribution $\mathbf{p}^*$, and (3) scales of entropy of the teacher's predictions. The latter two terms dominate in the large $N$ regime because the first term can be bounded by the Cauchy-Schwartz inequality and is of order $O(1/N)$. The second term quantifies how close the teacher is to the true distribution. The third term quantifies the teacher's uncertainty. A well-calibrated and certain teacher yields improved bounds on the generalization error of the student.

### 4.1.2 PROXY TEACHER

In order to lower the population risk and the generalization error of the student model, we introduce PTLoss that implicitly transforms the teacher's predictions to a *Proxy Teacher* under the KL loss. Then we leverage the calculated proxy teacher to guide the search of the perturbation coefficients that can minimize the right hand side of Eq. 16. Since the KL loss on the original teacher's predictions is a special case of PTLoss with all perturbed coefficients to be zero, we can improve the student by searching for the optimal coefficients for perturbation.

Concretely, we replace the loss function with a perturbed form as

$$\ell_{PT}(\mathbf{y}, \mathbf{f}(x)) = \ell(\mathbf{y}, \mathbf{f}(x)) + \mathbf{y}^T \sum_{m \in [M]} \eta_m \odot \left(\mathbf{1} - \frac{e^{\mathbf{f}(x)}}{Z(\mathbf{f}(x))}\right)^m, \quad (17)$$

where $\odot$ denotes element-wise multiplication between two vectors and $\eta_m = (\epsilon_m, \gamma_m) \in \mathbb{R}^2$ are hyperparameters for the perturbation at order $m$.

Similarly, we define $\boldsymbol{l}_{PT}(\mathbf{f}(x)) = \boldsymbol{l}(\mathbf{f}(x)) + \sum_{m \in [M]} \eta_m \odot \left(\mathbf{1} - \frac{e^{\mathbf{f}(x)}}{Z(\mathbf{f}(x))}\right)^m$.

Replace the distillation loss by the perturbed loss, we have

$$\tilde{R}_{PT}(\mathbf{p}^t, \mathbf{f}; \mathcal{S}) = \frac{1}{N} \sum_{n \in [N]} \left(\mathbf{p}^t(x_n)^T \boldsymbol{l}_{PT}(\mathbf{f}(x_n)) + \mathbf{p}^t(x_n)^T \log(\mathbf{p}^t(x_n))\right). \quad (18)$$

We first solve a Proxy Teacher $\mathbf{p}^{t_{PT}}$ such that $\tilde{R}(\mathbf{p}^{t_{PT}}, \mathbf{f}; \mathcal{S})$ is close to $\tilde{R}_{PT}(\mathbf{p}^t, \mathbf{f}; \mathcal{S})$. Then we find the optimal perturbation coefficients that yield the best risk bound.

Concretely, we define a transformation function $g : [0, 1]^2 \mapsto [0, 1]^2$ such that

$$\mathbf{p}^{t_{PT}}(x)^T \boldsymbol{l}(\mathbf{f}(x)) + \mathbf{p}^{t_{PT}}(x)^T \log(\mathbf{p}^{t_{PT}}(x)) = \mathbf{p}^t(x)^T \boldsymbol{l}_{PT}(\mathbf{f}(x)) + \mathbf{p}^t(x)^T \log(\mathbf{p}^t(x)), \quad (19)$$

where $\mathbf{p}^{t_{PT}}(x) = g(\mathbf{p}^t(x))$ is the Proxy Teacher. We transform teacher's predictions as $\mathbf{p}^{t_{PT}}(x) = g\left(\mathbf{p}^t(x)\right)$ so that the distilled risk $\tilde{R}_{PT}(\mathbf{f}; \mathcal{S})$ with the original teacher's predictions $\mathbf{f}^t(x)$ is close to $\tilde{R}(\mathbf{p}^t, \mathbf{f}; \mathcal{S})$ with the proxy teacher's predictions $\mathbf{p}^{t_{PT}}(x)$. In practice, $\mathbf{f}(x)$ is unknown, but we know the minimizer of the left hand side in Eq. 19 is $\mathbf{f}^{t_{PT}}(x)$ with softmax($\mathbf{f}^{t_{PT}}(x)$) = $\mathbf{p}^{t_{PT}}(x)$. Assume the student functional space is complicated enough, then we can replace the $\mathbf{f}$ by $\mathbf{f}^{t_{PT}}(x)$ and solve the above equation by numerical method.

---

[2]Note that the boundedness assumption on the loss is standard (Boucheron et al. (2005), Theorem 4.1; Menon et al. (2021), Proposition 2)

### 4.1.3 BEST PROXY TEACHER

For each set of $\eta_{[M]} = \{\eta_m | m \in [M]\}$, we can compute the risk deviation upper bound according to Proposition 1 omitting the $O(1/N)$ variance term:

$$D(\eta_{[M]}) = \left(\mathbb{E}_x \left[\|\mathbf{p}^{t_{PT}}(x) - \mathbf{p}^*(x)\|_2\right]\right)^2 + \mathbb{E}_x \left[\left(\mathbf{p}^{t_{PT}}(x)^T \log \mathbf{p}^{t_{PT}}(x)\right)^2\right]. \tag{20}$$

To compute the above deviation, we replace expectation by the sample mean. We also replace $\mathbf{p}^*(x_n)$ with $\mathbf{y}_n$ because it is an unbiased estimate of $\mathbf{p}^*(x_n)$.

Then we can minimize the empirical deviation:

$$\hat{D}(\eta_{[M]}) = \left(\frac{1}{N} \sum_{n \in [N]} \left[\|\mathbf{p}^{t_{PT}}(x_n) - \mathbf{y}_n\|_2\right]\right)^2 + \frac{1}{N} \sum_{n \in [N]} \left[\left(\mathbf{p}^{t_{PT}}(x_n)^T \log \mathbf{p}^{t_{PT}}(x_n)\right)^2\right]. \tag{21}$$

The first term is the deviation between the equivalent teacher and the ground truth, which encourages the teacher to be well-calibrated. The second term encourages the equivalent teacher to be as certain as possible. We randomly generate 100 sets of $\eta_{[M]}$ and pick the optimal $\eta_{[M]}^*$ that minimizes $\hat{D}$.

## 4.2 CONNECTION TO OTHER PERTURBATION METHODS

We compare PTLoss with the label smoothing method and claim that label smoothing proposed in (Szegedy et al., 2016) is a special case of PTLoss. Per the implementation in Szegedy et al. (2016), we can smooth the teacher labels in KD by

$$\mathbf{p}_0^{t_{ls}}(x_n) = (1 - \delta)\mathbf{p}_0^t(x_n) + \delta/2, \tag{22}$$

with a smoothing parameter $\delta$. Starting from Eq. 11, we can replace the term $\mathbf{p}_0^t(x_n)$ by its smooth version $\mathbf{p}_0^{t_{ls}}(x_n)$ and use $\mathbf{p}_1^t(x_n) = 1 - \mathbf{p}_0^t(x_n)$, which holds for binary classification. Then the original Eq. 11 with label smoothing is:

$$\ell_{KL}^{ls}\left(\mathbf{p}^t(x_n), \mathbf{p}^s(x_n)\right) \approx -\mathbb{H}\left(\mathbf{p}^{t_{ls}}(x_n)\right) + \mathbf{p}_0^{t_{ls}}(x_n)\left[-\log \mathbf{p}_0^s(x_n)\right] + \mathbf{p}_1^{t_{ls}}(x_n)\left[-\log \mathbf{p}_1^s(x_n)\right]$$
$$+ \mathbf{p}_0^{t_{ls}}(x_n)\left[\sum_{m=1}^{\infty} \epsilon_m (1 - \mathbf{p}_0^s(x_n))^m\right] + (1 - \mathbf{p}_0^{t_{ls}}(x_n))\left[\sum_{m=1}^{\infty} \gamma_m (\mathbf{p}_0^s(x_n))^m\right]. \tag{23}$$

For the entropy of the teacher output, the smooth version $\mathbb{H}\left(\mathbf{p}^{t_{ls}}(x_n)\right)$ is different from the original $\mathbb{H}\left(\mathbf{p}^t(x_n)\right)$ with only a constant $C$. We introduce $\Delta\mathbf{p}_0^t(x_n) = \delta/2 - \delta\mathbf{p}_0^t(x_n)$ and replace all the $\mathbf{p}_0^{t_{ls}}$ in Eq. 23 by $\mathbf{p}_0^{t_{ls}}(x_n) = \mathbf{p}_0^t(x_n) + \Delta\mathbf{p}_0^t(x_n)$, then we get:

$$\ell_{KL}^{ls}\left(\mathbf{p}^t(x_n), \mathbf{p}^s(x_n)\right) = \ell_{KL}(\mathbf{p}^t(x_n), \mathbf{p}^s(x_n))$$
$$+ \Delta\mathbf{p}_0^t(x_n)[-\log \mathbf{p}_0^s(x_n)] + (-\Delta\mathbf{p}_0^t(x_n))\left[-\log(1 - \mathbf{p}_0^s(x_n))\right]$$
$$+ \Delta\mathbf{p}_0^t(x_n)\left[\sum_{m=1}^{\infty} \epsilon_m (1 - \mathbf{p}_0^s(x_n))^m\right] + (-\Delta\mathbf{p}_0^t(x_n))\left[\sum_{m=1}^{\infty} \gamma_m (\mathbf{p}_0^s(x_n))^m\right]. \tag{24}$$

Again, by replacing the term $\log \mathbf{p}_0^s(x_n)$ and $\log\left(1 - \mathbf{p}_0^s(x_n)\right)$ with Eqs. 7 and 9, we have:

$$\ell_{KL}^{ls}\left(\mathbf{p}^t(x_n), \mathbf{p}^s(x_n)\right) = \ell_{KL}(\mathbf{p}^t(x_n), \mathbf{p}^s(x_n))$$
$$+ \Delta\mathbf{p}_0^t(x_n)[\sum_{m=1}^{\infty} \frac{1}{m}(1 - \mathbf{p}_0^s(x_n))^m] + (-\Delta\mathbf{p}_0^t(x_n))[\sum_{m=1}^{\infty} \frac{1}{m}(\mathbf{p}_0^s(x_n))^m]$$
$$+ \Delta\mathbf{p}_0^t(x_n)\left[\sum_{m=1}^{\infty} \epsilon_m (1 - \mathbf{p}_0^s(x_n))^m\right] + (-\Delta\mathbf{p}_0^t(x_n))\left[\sum_{m=1}^{\infty} \gamma_m (\mathbf{p}_0^s(x_n))^m\right]. \tag{25}$$

Thus, by setting

$$\epsilon_m^{ls} = \frac{\Delta\mathbf{p}_0^t(x_n)}{\mathbf{p}_0^t(x_n)}(\frac{1}{m} + \epsilon_m), \quad \gamma^{ls} = \frac{\Delta\mathbf{p}_0^t(x_n)}{\mathbf{p}_0^t(x_n)}(\frac{1}{m} + \gamma), \tag{26}$$

we get the same form as Eq. 12:

$$\ell_{KL}^{ls}\left(\mathbf{p}^t(x_n), \mathbf{p}^s(x_n)\right) \doteq \ell_{KL} + \mathbf{p}_0^t(x_n) \sum_{m=1}^{\infty} \epsilon_m^{ls} (1 - \mathbf{p}_0^s(x_n))^m + \left(1 - \mathbf{p}_0^t(x_n)\right) \sum_{m=1}^{\infty} \gamma_m^{ls} (\mathbf{p}_0^s(x_n))^m. \tag{27}$$

| Dataset | Task | Train | Distillation | Dev | Test |
|---------|------|-------|--------------|-----|------|
| MNLI | Natural Language Inference | 58,905 | 314,161 | 19,636 | 9,832 |
| SST-2 | Sentiment Analysis | 6,734 | 53,870 | 6,736 | 872 |
| BoolQ | Boolean Question Answering | 2,500 | 5,927 | 1,000 | 3,270 |

Table 1: Dataset Statistics

## 5 EXPERIMENTS

**Tasks and Datasets.** We conduct experiments on three public benchmark datasets, including MNLI Williams et al. (2017) for multi-genre natural language inference, SST-2 (Wang et al., 2018) for sentiment analysis, and BoolQ (Clark et al., 2019) for boolean question answering. The dataset statistics are shown in Table 1.

**Model Architecture.** For the teacher model, we choose the T5 architecture (Raffel et al., 2020) and select three teacher models of different scales. Specifically, we use T5-xxl with 11 billion parameters, T5-xl with 3 billion parameters, and T5-large with 770 million parameters. For the student model, we use BERT-base model (Devlin et al., 2018) with 110 million parameters.

**Compared Methods.** We compare PTLoss with the following baselines: 1) Standard KL loss (Kullback, 1959): use standard KL loss in knowledge distillation; 2) Temperature scaling (Hinton et al., 2015): scale the model logits via a temperature hyper-parameter; 3) Label smoothing (Szegedy et al., 2016): smooth the teacher's output by a small scalar; 4) Focal loss (Lin et al., 2017): modulated cross-entropy loss to focus learning on hard examples.

### 5.1 MAIN RESULTS

| Dataset | Teacher Size | Teacher Acc. | KL | Temp. | Smoothing | Focal | PTLoss |
|---------|--------------|--------------|-----|-------|-----------|-------|--------|
| MNLI | T5-xxl | 94.68 | 90.21 | 90.79 | 90.15 | 90.30 | **90.94** |
| | T5-xl | 92.42 | 90.41 | 90.32 | 88.94 | 88.83 | **90.84** |
| | T5-large | 93.56 | 89.95 | 90.36 | 90.46 | 90.42 | **90.62** |
| SST-2 | T5-xxl | 96.44 | 88.88 | 89.56 | 89.56 | 90.14 | **90.25** |
| | T5-xl | 95.18 | 89.67 | 89.68 | 90.02 | 89.22 | **90.25** |
| | T5-large | 95.53 | 88.89 | 89.56 | 89.56 | 89.45 | **90.02** |
| BoolQ | T5-xxl | 89.14 | 69.57 | 72.23 | 68.38 | 68.78 | **72.69** |
| | T5-xl | 87.52 | 70.40 | 72.66 | 68.10 | 69.51 | **72.87** |
| | T5-large | 77.91 | 69.39 | 70.03 | 69.39 | 69.27 | **70.83** |

Table 2: Main Results on three datasets. The student model is distilled from teacher models with different size. We show student model's test accuracy (%) and list the teacher model's validation accuracy (%) in the colored column for reference. The details of the hyper-parameter search for each method are introduced in Appendix A.1.

Table 2 shows the performance of the PTLoss and the baselines. In this set of experiments, we set the perturbation order as 5 in the PTLoss, the corresponding perturbation coefficients are obtained through our Proxy Teacher method. We found that PTLoss outperform all the baselines under 9 settings, the average performance improvement to the standard KL is 1.31%. We found that for the most challenging task BoolQ, PTLoss presents the most prominent improvement to the underlying standard KL loss.

Among all the baselines, temperature scaling performs strongly on MNLI and BoolQ, while label smoothing and focal loss yield competitive results on SST-2, but the PTLoss holds a lead to the strongest baseline method on all the tasks. Besides, we present different-sized teacher models on each dataset. The results show that the performance gain is consistent regardless of the capacity of teacher models. Notably, we report the baseline results with an exhaustive hyper-parameter search as shown in Appendix A.1. Instead, we only give the perturbation order for PTLoss and obtain the perturbation coefficients via the Proxy Teacher method. Although the hyper-parameter search space

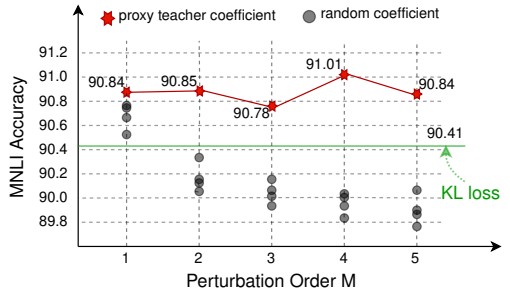 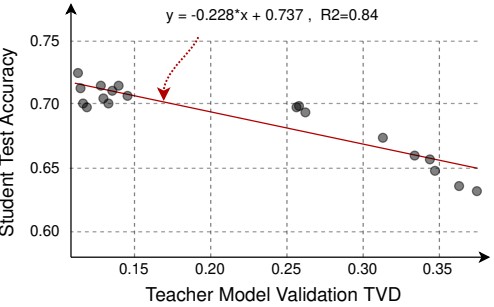

(a) The Proxy Teacher method v.s. random search for the perturbation coefficient selection. We conduct experiments on MNLI with a T5-xl teacher.

(b) Correlation between the validation TVD of the teacher model and the test accuracy of the student model. Experiments are conducted on BoolQ.

Figure 2: PTLoss analysis.

of PTLoss is much larger than the baselines, we still save much human labor owing to the automatic process of Proxy Teacher method compared to the baselines.

## 5.2 PERTURBATION COEFFICIENTS SEARCH VIA THE PROXY TEACHER

We validate the effectiveness of the Proxy Teacher method for perturbation coefficients selection. We use MNLI as the representative dataset to show the difference between the Proxy Teacher and the random search method. For each perturbation order, Proxy Teacher follows the procedure as in Section 4.1.3 to select the coefficients on the development set by minimizing the empirical deviation as shown in Eq. 21, while random search just samples a $M$-dimension vector with each perturbation coefficient between $[-1, 10]$.

In Fig. 2a, we range the perturbation order ranged from 1 to 5 and report the student model accuracy with different perturbation coefficients obtained by either the Proxy Teacher or random search. The consistent improvement over the random coefficient demonstrates the importance of adding appropriate perturbation via a systematic coefficients search. If we just randomly set the perturbation coefficients, the student performance can drop by up to $1.2\%$. Comparing different perturbation orders, we also found that the higher the perturbation order, the higher performance difference between the Proxy Teacher and the random coefficients. This is because in the higher-dimension space, it is harder for random search to get a set of appropriate perturbation coefficients, which makes the random PTLoss even worse than the standard KL loss. Instead, equipped with the perturbation coefficients obtained via Proxy Teacher PTLoss significantly outperforms the underlying KL loss.

We limit the perturbation order in practice because when the order is high, the corresponding perturbation term will be a low value after the power operation, compared to the leading terms. Also, a too high perturbation order will make it difficult to search for the optimal perturbation coefficients.

## 5.3 CORRELATION BETWEEN THE TEACHER MODEL'S TVD AND THE STUDENT MODEL PERFORMANCE

Fig. 2b presents the student model performance positively correlated with the total variance distance between the teacher model's output and the ground truth. By calculating the validation TVD of the teacher model, we measure the discrepancy between the teacher's output and the ground truth data. The results demonstrate that the teacher model with higher-fidelity predictive distribution yields a better distilled student.

## 6 CONCLUSION

We proposed PTLoss to perturb the teacher model's output distribution to a high-fidelity one for student model training in knowledge distillation, followed by the Proxy Teacher method to systematically search perturbation coefficients by calculating the implicitly shifted teacher. Moreover, we theoretically established a bounded distillation risk of the proposed PTLoss and illustrated its advantage over the standard KL loss. We also demonstrated the other perturbation methods such as label smoothing fall into the special cases of PTLoss. The empirical study further supported our theory and validated the effectiveness of our method.

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

# A  APPENDIX

## A.1  HYPER-PARAMETERS

We list the search range of hyperparamters in Table 3. The search of batch size and learning rate is applied to all the methods. And for each baseline, we search for the best baseline-specific hyperparameters.

| Hyper-parameter | Search Range |
|---|---|
| Learning Rate | $\{2, 3, 5\} \times 10^{-5}$ |
| Batch Size | $\{8, 16, 32, 64, 128, 256\}$ |
| Temperature $T$ | $\{0.1, 0.2, 0.5, 1.0, 2.0, 5.0, 10\}$ |
| Label Smoothing $\delta$ | $\{0.02, 0.05, 0.1, 0.15, 0.2\}$ |
| Focal Loss $\tau$ | $\{0.1, 0.2, 0.5, 1, 2.0, 5.0\}$ |
| Random PTLoss $\epsilon_j$ | $[-1, 10]$ |
| Random PTLoss $\gamma_j$ | $[-1, 10]$ |

Table 3: The search range of hyper-parameters.

## A.2

Running time For each perturbation order, we randomly sample 100 coefficients from $[-1, 10]$. Then we compute and choose the best scores according to Eq. 21 on the dev dataset with 1000 examples. The whole process takes less than two minutes on CPU with 64G memory.

## A.3  PERTURBATION COEFFICIENT

Figure 3 shows a parameter study of the perturbation coefficients $\epsilon$ and $\gamma$. We adopt 1-order PTLoss and search $\epsilon$ and $\gamma$ in $[-1, 5]$ to show how the perturbation extent determined by the perturbation coefficients.

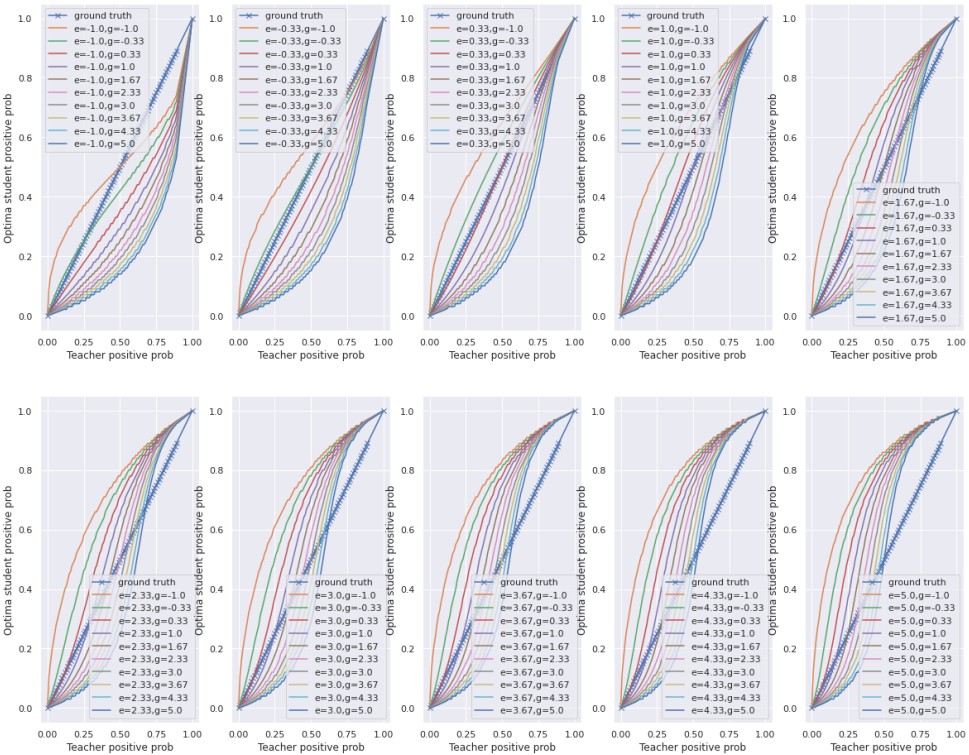

Figure 3: Parameter study of PTLoss. We fix the perturbation order and varies the coefficients.

## A.4 PROOF OF PROPOSITION 1

*Proof.* By Eqs. 5 and 15,

$$\tilde{R}(\mathbf{p}^t, \mathbf{f}; \mathcal{S}) - R(\mathbf{f}) = \frac{1}{N} \sum_{n \in [N]} \mathbf{p}^t(x_n)^T \boldsymbol{l}(\mathbf{f}(x_n) + \frac{1}{N} \sum_{n \in [N]} \mathbf{p}^t(x_n)^T \log(\mathbf{p}^t(x_n))) - \mathbb{E}_x[\mathbf{p}^*(x)^T \boldsymbol{l}(\mathbf{f}(x))].$$

(28)

Let

$$\Delta = \frac{1}{N} \sum_{n \in [N]} \mathbf{p}^t(x_n)^T \boldsymbol{l}(\mathbf{f}(x_n) - \mathbb{E}_x[\mathbf{p}^*(x)^T \boldsymbol{l}(\mathbf{f}(x))],$$

and

$$H = \frac{1}{N} \sum_{n \in [N]} \mathbf{p}^t(x_n)^T \log(\mathbf{p}^t(x_n))),$$

then

$$\mathbb{E}\left[(\tilde{R}(\mathbf{p}^t, \mathbf{f}; \mathcal{S}) - R(\mathbf{f}))^2\right] = \mathbb{E}\left[(\Delta + H)^2\right]$$
$$\leq 2\mathbb{E}\left[\Delta^2\right] + 2\mathbb{E}\left[H^2\right]$$
$$= 2\mathbb{V}\left[\Delta\right] + 2\mathbb{E}\left[\Delta\right]^2 + 2\mathbb{E}\left[H^2\right]$$

where the second line is by the inequality $(a+b)^2 \leq 2a^2 + 2b^2$ and linearity of expectation, and the third line is by $\mathbb{E}\left[\Delta^2\right] = \mathbb{V}\left[\Delta\right] + \mathbb{E}\left[\Delta\right]^2$. Observe that

$$\mathbb{E}\left[\Delta\right] = \mathbb{E}_x\left[(\mathbf{p}^t(x) - \mathbf{p}^*(x))^T \boldsymbol{l}(\mathbf{f}(x))\right]$$
$$\leq \mathbb{E}_x\left[\|\mathbf{p}^t(x) - \mathbf{p}^*(x)\|_2 \cdot \|\boldsymbol{l}(\mathbf{f}(x))\|_2\right]$$
$$\leq \mathbb{E}_x\left[\|\mathbf{p}^t(x) - \mathbf{p}^*(x)\|_2 \cdot c_1 \cdot \|\boldsymbol{l}(\mathbf{f}(x))\|_\infty\right]$$
$$\leq c_2 \mathbb{E}_x\left[\|\mathbf{p}^t(x) - \mathbf{p}^*(x)\|_2\right],$$

where the second line is by the Cauchy-Schwartz inequality, the third line by the equivalence of norms with a constant $c_1$, and the last line is by the boundedness of loss term.

Since $R(\mathbf{f})$ is a constant,

$$\mathbb{V}\left[\Delta\right] = \mathbb{V}\left[\tilde{R}(\mathbf{p}^t, \mathbf{f}; \mathcal{S})\right] = \frac{1}{N} \cdot \mathbb{V}\left[\mathbf{p}^t(x)^T \boldsymbol{l}(\mathbf{f}(x))\right].$$

By plugging in everything above, we finish the proof. $\qquad\square$

