# OpenReview forum: "Do Not Blindly Imitate the Teacher: Loss Perturbation for Knowledge Distillation"
_ICLR.cc/2023/Conference — Submitted to ICLR 2023_

### Official Review · Reviewer_Djov · 2022-10-16

**Confidence:** 4
**Correctness:** 3
**Technical Novelty And Significance:** 3
**Empirical Novelty And Significance:** 3
**Recommendation:** 6

**Clarity, Quality, Novelty And Reproducibility:**

I think the paper has some novelty. But in terms of clarity, quality and reproducibility, many improvements are needed as I have listed above.

**Details Of Ethics Concerns:**

No ethical concerns.

**Strength And Weaknesses:**

Strengths:

1. The authors consider an interesting modification to the usual KL objective by perturbing the leading terms in its Maclaurin expansion for knowledge distillation.

2. They show how this perturbation subsumes label smoothing as a special case.

3. Experimental results on multiple NLP datasets seem to show that the proposed objective outperforms the usual KL loss, LS loss and others.

Weaknesses and suggestions for improvement:

I think there are quite a few weaknesses and things that are unclear in this version of the paper, I list them below:

1. Based on my reading of the paper, it appears that only the KL loss, LS loss or the proposed PTLoss is used for knowledge distillation. It is my understanding that a combination of cross-entropy loss and a version of the KL loss is used usually. I am wondering if the authors missed mentioning this in the paper or if they have a different setup.

2. Definition of true risk $R(\mathbf{f})$ is defined differently in Eq. (1) and in Eq. (28) in the supplement. I think Eq. (28) makes more sense in that there is a true distribution $\mathbf{p}^*(x)$ that is inaccessible and we are trying to approximate with the PTLoss and the proxy teacher. If $\mathbf{y}$ (the one-hot labels) were the true distribution, then we would use it directly.

3. It is not at all clear from the paper as to how the best Proxy Teacher is found in Eq. (19) and how that relates to Eq. (21) to find the best $\eta$. For Eq. (19), the authors just say that it is solved numerically, perhaps they should elaborate. My guess is that random $\eta$'s are sampled, and for each $\eta$, $\mathbf{p}^{t_{PT}}$ is found numerically and the corresponding ${\hat{D}}$ in Eq. (21) is computed. Finally, the $\eta$ that corresponds to the $\mathbf{p}^{t_{PT}}$ that results in the smallest ${\hat{D}}$ is selected. The authors should clarify whether this is the procedure, and should make this clear in the paper.

4. There are parts of in Section 4.1.2 and 4.1.3 that are seem to be wrong or assumptions that are not justified. The authors write towards the end of 4.1.2 that "Assume the student functional space is complicated enough, then we can replace the $\mathbf{f}$ by $\mathbf{f}^{t_{PT}}(x)$. I don't understand what this means or how we can do this.

In Section 4.1.3, the authors write: "We also replace $\mathbf{p}^*(x_n)$ with $\mathbf{y}_n$." This does not make a lot of sense to me. If the whole point is to get closer to $\mathbf{p}^*(x)$, why replace it with one-hot labels. If one-hot labels were good, using just the cross-entropy loss should be fine right? l think this is the main problem with the theoretical analysis.

5. All the experiments are done with real-world datasets with unknown $\mathbf{p}^*(x)$. What the paper needs is an illustrative experiment with a toy dataset where the $\mathbf{p}^*(x)$ is known exactly, so that the authors can show that the proposed PTLoss gets closer to the desired output compared to one-hot labels, KL loss and LS loss. I would suggest the authors look at Section 2.2 in this recent paper which does exactly this for essentially the same problem:

Rin et al. BETTER SUPERVISORY SIGNALS BY OBSERVING LEARNING PATHS, ICLR 2022

I think this is an important and necessary experiment

6. All the theoretical analysis presented is for binary classification. Can the authors provide details how to extend these ideas for multi-class problems?

7. Experimental results should be reported over multiple runs, with means, standard deviations and statistical significance results. Otherwise, it is not possible to determine whether the performance improvements exist.

8. I would suggest the authors discuss the FilterKD that I mentioned above. I think the ideas are very closely related. Ideally, experimental comparisons should be provided.

9. Similarly, I would like the authors to present some comparisons against SOTA KD methods such as Contrastive Representation Distillation which I believe would help readers a lot, but this is not required.

10. I am not very familiar with the datasets used in the paper. It would help me a lot and many readers I think if the authors could consider an image recognition dataset such as CIFAR-100 or ImageNet-100. Again, this is just a suggestion, not too important.

11. This is minor, but using $[0,1]^2$ as the support for a probability distribution is not perfect, as not all pairs of numbers in the $[0,1]$ interval would form a probability vector.

12. I think it should be $\mathbf{y}^T log(Z(f(x)))$ in Eq. (2) and there should be a $\mathbf{y}^T$ multiplying the RHS is Eq. (4) right?

**Summary Of The Paper:**

KL divergence between the teacher outputs and the student outputs is commonly used for knowledge distillation. However, this may not be optimal. The authors propose a method to perturb the KL divergence by modifying terms in its Maclaurin expansion. This implicitly modifies the teacher outputs to form a "Proxy Teacher". The authors claim and derive a proof that the proxy teacher has the ability to reduce the gap between the true risk and the distillation risk. Connection to Label Smoothing (LS) is also derived. Experiments on multiple NLP datasets show improved performance from the proposed PTLoss over the usual KL loss as well as related methods like LS.

**Summary Of The Review:**

I think the authors are tackling an important problem, and they have an interesting idea to improve the usual KL loss for knowledge distillation. However, I don't think the theoretical analysis is sound and I believe the experiments needs to be done better as well. Given these concerns, I am recommending rejection.

I hope the authors can respond to my review in case I have misunderstood some parts of the paper.

UPDATE AFTER AUTHORS' RESPONSE:

The authors use a KD setup which is different from the usual setup of having access to both ground-truth labels and teacher outputs -- only the teacher outputs are used for KD in this paper. I think the theoretical analysis makes more sense to me now. However, having experiments which also use the cross entropy loss assuming known ground-truth can help make the paper more broadly useful. And also including an experiment on CIFAR-100 and/or Tiny ImageNet to make it easier to put this work in context with the dozens of existing KD works. I do appreciate the results on the toy dataset which seems to agree with their theoretical analysis. Given these improvements to the paper, I am increasing my rating to a 6.

---

> ### Author Response · Authors · 2022-11-19
> **Author Response to Reviewer Djov - Part 1**
>
> We thank reviewer Djov for the thorough review.
>
>  For some clarification questions, we have modified our draft to address them, we hope the updated version can be more reader-friendly. Below is our response.
>
> 1. Compared to the classical KD loss, which is a combination of cross-entropy loss and KL loss, we have a different setup. The cross-entropy loss computes the loss between the student outputs and the ground truth data, while the KL loss computes the loss between the student outputs and the teacher outputs. In our setting, we do not have the ground truth data and can only train the student from the teacher’s output.
>
> This setting is reasonable in practice because the teacher models (or only their APIs) are sometimes provided alone. This is either because the model is too large to be directly distributed/finetuned or to prevent some potential misuse of it like what the GPT-3 initially did, or to prevent potential training data leakage.
>
> 2. The definition of the true risk $R(\mathbf{f})$ in Eq. (28) is derived from Eq. (1). There is an Eq. (15) in the paper, which is a little bit far away from either Eq. (1) or Eq. (28),  showing how we start from Eq. (1) and write it in a Bayes classifier form. This formulation can be found in Menon et al. A Statistical Perspective on Distillation, ICML 2021.
>
> 3. We appreciate the reviewer's deep thinking about this part, we omitted this part due to the page limit in the draft.
> The speculated process is exactly correct. We randomly sample $\eta$ (which is changed to $\epsilon_{[M]}$ in the latest version) and use Newton-Raphson method to compute the $\mathbf{p}^{t_{[PT]}}$ numerically. For each $\epsilon_{[M]}$, we can compute the empirical deviation as the Eq.(21) shows, and we generate 100 sets of $\epsilon_{[M]}$ and pick the optimal one that minimizes $\hat{D}$.
>
> 4. For Sec 4.1.2, $f(x)$ is the student's logits given the input text x. $f^{t_{PT}}(x)$ is the equivalent teacher's logits given the input text $x$.
> Ideally, we want to learn the transformation function g such that for any function $f$, the Eq. (19) holds. However, the system is not solvable. But we can possibly solve the equation for a given function $f$. Then the question becomes which f to choose. In practice we care the most on the optimal student model that minimize the distillation risk (Eq. (18)), which is $f^{t_{PT}}$ upon solving the Eq. (19). Thus we replace $f$ by $f^{t_{PT}}$ in Eq. (19) to solve $g$.
>
> For Sec. 4.1.3, we are trying to find a set of perturbation parameters $\epsilon_{[M]}$,  once the optimal $\epsilon_{[M]}$ is obtained, we proceed to the student model training stage (distillation stage).
> To find the minimizer of Eq.(20), we need an unbiased estimator of  $\mathbf{p}(x_n)^*$, so we replace it by $\mathbf{y}_n$.
> The $\mathbf{y}_n$ comes from a hold-out dataset for searching the perturbation parameters.
> For the distillation stage, we do not have the ground truth label of the distillation set, so the student model is trained using the teacher's soft output.
>
> 5. Thanks for this point, it is a really helpful suggestion and we do conduct experiments on a synthetic dataset following the FilterKD paper. The results are shown in Sec. A.7. We generate a Gaussian dataset with 3 classes and $10^5$ samples. In Fig. 4(a), we plot the L2-distance and the accuracy on the test set. Among the baselines including labeling smoothing (LS) and knowledge distillation with standard KL (KD), we observed that PTLoss can achieve smaller L2-distance and higher accuracy.
>
> In Fig.4(b), we further study the correlation between the L2-distance and the test accuracy of differently perturbed teachers. Specifically, we sampled 10 perturbed teachers in different stages of the best proxy teacher searching process and compare their results. The fitting curve reveals that the smaller the L2-distance, the better the corresponding model. Actually, in our original draft, we have similar results on a real-world dataset (Fig.2 (b)). In that set of experiments, we computed the TVD of teacher models using a validation set as the discrepancy measurement between the teacher's output distribution and the ground truth data. The results demonstrate that the teacher model with high-fidelity predictive distribution can yield a better distilled student.
>
> 6. For brevity, we used binary classification in our previous derivation of the paper. In the updated draft, we have extended it to the multi-class setting. The changes can be found in Sec.4. Correspondingly, we added a multi-class dataset ANLI to validate the effectiveness of PTLoss in Sec. A.6. The result shows PTLoss can bring significant improvement compared to its underlying method in this setting.

---

> ### Author Response · Authors · 2022-11-19
> **Author Response to Reviewer Djov - Part 2**
>
> 7. The multiple-run results are reported in Table 4. In our previous implementation, for each baseline method, we did a grid search for the hyperparameters as Appendix A.1 shows and we listed the best performance after the parameter search. Now with the multiple-run results, we found that PTLoss still outperforms the baselines consistently with a statistically significant margin.
>
> 8. Thanks for your suggestion. We incorporate FilterKD in our baselines. The comparison can be found in Table 4.
>
> 9 - 10. Additionally, we incorporate Flooding (Ishida et al. Do We Need Zero Training Loss After Achieving Zero Training Error?, ICML 2020.) in the baselines, which is a general regularization method to intentionally prevents further reduction of the training loss when it reaches a reasonably small value.
>
> For the image-related tasks, we noticed that reviewers showed interest without exception. Due to the time limit, however, we just humbly claim the effectiveness of our method in text-related tasks, and we welcome other researchers with more experience in the CV field can explore the potential application of PTLoss.
>
> 11. We assume the reviewer is talking about the transformation function $g$ in Eq. (19). Because the $\mathbf{p}^t(x)$ and $\mathbf{p}^{t_{PT}}(x)$ is probability vector, so the definition of $g$ is relatively relaxed.
>
> 12. In Eq.(2) and Eq.(4), it involves some mathematical transformation. Specifically, in Eq.(2)  we use the expression of $\log{\mathbf{p}(x)}$ and the fact that $\mathbf{y}^T\mathbf{1} = 1$, thus there is no $\mathbf{y}$ involved in the last term of the above equation. We have added more context in the updated draft to make it easier to understand.

---

> ### Author Response · Authors · 2022-12-04
> **Follow-up Response to Reviewer Djov**
>
> Dear reviewer Djov, thank you for recognizing our improvement in the revised paper draft and for your further suggestions.
>
> For the KD setup used in our paper, we will add more context in the final version to illustrate its significance and practicality -- with the prevalence of large pre-trained models and the pursuit of less annotation, as well as the data privacy requirements for model training pipelines in industrial scenarios, our setting, in which only the teacher outputs are available, is becoming increasingly reasonable. Meanwhile, we will incorporate your suggested experiments in the final version, where the ground-truth distillation set is also available, to make the paper more broadly useful.
>
> For the experiments on image datasets, the time is limited for us to present comprehensive results during the rebuttal stage, but we are willing to try our best to show the effectiveness of our method on these tasks in the final version.
>
> Again, we appreciate your raised rating and your feedback on our response.

---

### Official Review · Reviewer_BGUG · 2022-10-16

**Confidence:** 4
**Correctness:** 3
**Technical Novelty And Significance:** 2
**Empirical Novelty And Significance:** 2
**Recommendation:** 3

**Clarity, Quality, Novelty And Reproducibility:**

Clarity is good.

Quality and Novelty are fair.

Reproducibility is unknown.

**Strength And Weaknesses:**

Pros.

1. Both empirical and theoretical analyses are provided.
2. The paper is well-written and easy to follow.


Cons.

1. Related works are outdated and not sufficient. Take knowledge distillation as an example. There are tens of KD papers from 2021 and 2022, but none of them are cited.
2. Only limited datasets are considered. The paper's proposals seem to be general methods, which should be validated in diverse representative tasks (e.g., ImageNet classification, detection, segmentation, and GLUE NLP benchmarks) to support their effectiveness.
3. The improvements are marginal. Need multiple runs and an error bar to show the significance of benefits.
4. Missing comparisons with important literature like "Revisiting Knowledge Distillation via Label Smoothing Regularization" and "Do We Need Zero Training Loss After Achieving Zero Training Error?".
5. The authors argue the limitation of a discrepancy between the teacher's output distribution and the ground truth label distribution. It is hard to say the discrepancy is a bad thing. Because lots of input samples like ImageNet images in vision can be multi-label but the ground truth is usually a one-hot label. Thus, a soft label is actually more appropriate to represent the true data distribution,

**Summary Of The Paper:**

Summary.

This paper is dedicated to investigating knowledge distillation. The authors argue that the vanilla learning objective of knowledge distillation is sub-optimal since there exists a discrepancy between the teacher's output distribution and the ground truth label distribution. Therefore, they introduce the PTLoss with a Maclaurin series and perturbed leading-order terms. Both theoretical and empirical results are conducted.

**Summary Of The Review:**

A borderline paper needs more studies.

---

> ### Author Response · Authors · 2022-11-19
> **Author Response to Reviewer BGUG - Part 1**
>
> The authors thank the reviewer for the careful review. Our point-by-point response is as follows:
> 1. We thank the reviewers for pointing out that more related works are needed. Because we focus on text-related scenarios, some KD papers, especially for those studying CV tasks, are not discussed in our draft. In the updated version, we have added the following papers:
> - Li et al. Revisiting Knowledge Distillation via Label Smoothing Regularization, CVPR 2020.
> - Ishida et al. Do We Need Zero Training Loss After Achieving Zero Training Error?, ICML 2020.
> - Tian et al. Contrastive Representation Distillation, ICLR 2020.
> - Rin et al. Better Supervisory Signal by Observing Learning Paths, ICLR 2022.
>
> And for the first two papers mentioned by reviewer BGUG, we discussed them in detail in bullet 4.
>
> 2. We theoretically and empirically demonstrate the effectiveness of our method. Due to the page limitations, we only present limited empirical studies. In the updated draft, we have supplemented the following settings to make the empirical study more comprehensive:
> - we show the proposed PTLoss can be extended to multi-class settings and compared it with its underlying standard KL loss on a multi-class classification task;
> - we use a synthetic Gaussian dataset, where the ground truth distribution is known, to further support our theoretical analysis.
> We hope the above content can help convince readers of the effectiveness of PTLoss.
>
> For the mentioned image-related tasks, we understand KD is broadly studied in the CV community and we welcome researchers to apply PTLoss in their interested fields. Currently, we simply confine the empirical study to the text-related tasks to sharpen our focus, but we are definitely willing to add more diverse tasks in later versions to make more impact.
>
> 3. First, for each baseline method, we do a grid search for the hyperparameters as Appendix A.1 shows and we list the best performance after the parameter search. For our method, we search for the perturbation coefficients by the proposed proxy teacher method as Eq. (19) shows. In practice, it is implemented by numerical method, and we use only 100 rounds for a 10-dimension vector  (5-order perturbation involves 10 coefficients) search to get the solution, this number of searching is very moderate compared to its dimension. If we add the search rounds, we can find better perturbation coefficients and further improve the performance.
>
> Second, we implement multiple runs and update the results as Table 4 in the updated draft shows. The results show our method consistently outperforms all the baselines and the improvements are statistically significant, and the improvement is especially prominent compared to its underlying method - the standard KL loss. Moreover, for the not-too-difficult tasks such as MNLI and SST-2, the baseline performance is already very high (~90%), so it does not make sense to pursue a large performance improvement in that case. In Table 5, we conduct experiments on a challenging task ANLI, where the baseline performance is only 44.22%, then the performance improvement is much more significant (3.45%).
>
> Third, though we admit the SOTA result is important for a proposed method, we agree with ICLR's reviewer guide, in which it is explicitly stated that a lack of SOTA results does not by itself constitute grounds for rejection. Not to mention that we show theoretically and experimentally that our PTLoss can achieve better performance compared to the baselines. We hope that the latest results and our response can mitigate the negative comments regarding the experiment part.

---

> ### Author Response · Authors · 2022-11-19
> **Author Response to Reviewer BGUG - Part 2**
>
> 4. We thank the reviewer for sharing these concrete literature with us.
>  - For the first work: Revisiting Knowledge Distillation via Label Smoothing Regularization(Tf-KD), we have an orthogonal setting to them. Tf-KD studies the scenario in which a strong teacher model is unavailable, and the resulting student models are worse than the normal KD in most cases. Even though there are several datasets that Tf-KD is better than the normal KD, the performance gain is very marginal that the largest improvement is <0.2%. This runs in the opposite direction to our setting: we have large pre-trained teacher models and we pursue small well-learned student models.
>
> Moreover, for the comparison between LS and KD, both of us try to seek the connection between LS and KD but we head in different directions. Tf-KD starts from the Re-KD and De-KD experiments and analyzes the mathematical form of the two loss functions. They summarize the relationship between KD and LS as KD is a special case of LS where the smoothing distribution is learned but not pre-defined. In Section 4.2 of our paper, we also present the connection between our PTLoss to the LS. Instead, we have totally different motivations and conclusions. We regard both methods as perturbation methods and proved that LS is a special case of PTLoss. This conclusion reveals that by leveraging the Maclaurin series and perturbing the polynomial terms, we are able to construct a complex perturbation space that includes other perturbation methods such as LS, thus PTLoss are more capable of approaching the ground truth distribution via perturbation than the other perturbation methods.
>
> - For the second work, Do We Need Zero Training Loss After Achieving Zero Training Error (Flooding), it is a general regularization method to intentionally prevents further reduction of the training loss when it reaches a reasonably small value. This work is motivated by the overfitting issues of the overparameterized deep networks, but its simple formulation fails to manipulate the supervision signal (i.e., the teacher outputs in the KD problem) to be closer to the ground truth distribution. Instead, we theoretically demonstrate the perturbed teacher can improve the student model by reducing the distillation risk, and we provide a systematical approach to add such perturbation. This method is more appropriate to KD problem, compared to the general regularization method.
>
> We include Flooding in our baselines, the results are shown in Table 4 of the updated draft. PTLoss presents a consistent empirical advantage to Flooding in all three settings.
>
> 5. The ground truth distribution $p*(x)$ is unknown, we are not referring it to the one hot label in our paper. Even if the teacher’s output distribution is closer to the p*(x) than the one-hot label, it is still a learned approximation of p*(x) and as we described in Sec 4.1, we can further use a proxy teacher that is closer to p*(x) to supervise the student model’s training.
>
> Besides, our experiment on both the real-world dataset and the synthetic dataset shows reducing the dependency between the teacher's output distribution and the (approximated) ground truth distribution is beneficial for student model training.  Fig.2 (b) and Fig.4 (b) present that when the teacher's output distribution is closer to the ground truth,  it can yield a better student model.

---

> ### Author Response · Authors · 2022-12-04
> **Does our response address your concerns?**
>
> Dear reviewer BGUG, thanks again for your thoughtful review.
>
> You may find our responses from both the general and individual posts. We are looking forward to your valuable feedback and we would appreciate the opportunity to engage further if needed.  It will be important to us if you can take our responses into account and consider raising your recommendation in the final assessment. Thank you in advance.

---

> ### Comment · Reviewer_BGUG · 2022-12-10
> **Response to the rebuttal**
>
> Many thanks for all the efforts during the rebuttal.
>
> 1. My first concern is partially addressed. If the authors only focus on the text-based KD. Please change the title in the related work to "text-based KD".
>
> 2. My second concern is not addressed. In the draft, from the abstract to the experiments, the authors advocate the general benefits of PTLoss compared to KD. To claim a general advantage over another general applicable approach KD, a comprehensive and diverse evaluation is needed.
>
> 3. My third concern is partially addressed. Thanks for the results of multiple runs. The concern of marginal improvements is highly related to a diverse evaluation mentioned in Cons 2.
>
> 4. For my 4th concern, Tf-KD is a special LS. The authors compare PTLoss to vanilla LS, why miss a more advanced LS - Tf-KD? I think a comparison is needed.
>
> 5. My last concern is addressed.
>
> Thanks. I remain my score unchanged. Suggestions: a more diverse and comprehensive evaluation with sufficient baselines or tune down the benefit claim of PTLoss.

---

> > ### Author Response · Authors · 2022-12-13
> > **Response to the follow-up comments from Reviewer BGUG**
> >
> > We appreciate the further engagement of Reviewer BGUG.
> > 1. Thank you for pointing this out, we will change the subsection title of the related work in the final camera-ready version.
> > 2. First, we are not excessively claiming PTLoss is better than KD. Instead, we are advocating that PTLoss is better than KL loss (and several other compared baseline loss designs) for KD. This difference is important in our setting where only the teacher models are provided.
> >
> > Second, in the first-round response, we have added experiments on a multi-class dataset and a synthetic Gaussian dataset, we have also included two more baselines FilterKD and Flooding in our main results.
> > Per the reviewer’s suggestion, we further conduct experiments on CIFAR-100. The results are as follows:
> > | Model | Teacher Acc. | Baseline | Tf-KD	| PTLoss |
> > | :--- | :----: | :----: | :----: | ----: |
> > | ResNet18 | 76.03 | 75.87 | 76.65 | **77.48** |
> > | GoogleNet | 78.31 | 78.72 | 79.64 | **80.22** |
> > | DenseNet121 | 79.04 | 79.04 | 79.58 | **80.12** |
> >
> > Here we follow the implementation of **Tf-KD_self**, which can be slightly modified to be adapted to our setting. We introduce the implementation details in bullet 4. Note: The baseline results are same as in Table 5 of the Tf-KD paper(Yuan et al., 2020). The Tf-KD results are implemented by us. It is different from the original paper because we disable the ground truth data during the distillation stage.
> >
> > 3. We add more datasets (multi-class MNLI, synthetic dataset, CIFAR-100) and more tasks (image classification) to show the effectiveness of our work. We present the results of the multi-class MNLI and the synthetic dataset in A.6 and A.7 in the updated paper draft. The CIFAR-100 results are shown in this response.
> >
> > 4. As we pointed out in bullet 4 of our previous response, Tf-KD has an orthogonal setting to our work. In Tf-KD, ground truth data is required to implement both Tf-KD_self and Tf-KD_reg, and they assume the teacher model is unavailable. In our setting, however, the ground truth data is unavailable, while the teacher model is provided alone. That’s why we do not initially include this comparison in our first-round response.
> > In this round, we adopt Tf-KD to our setting and include it in the baseline. To make a fair comparison, we disable the ground truth data in the distillation stage, and use a pre-trained student model as the teacher model for distillation. Specifically, we modified Eq.(7) in the Tf-KD paper as $L_{self} = D_{\text{KL}}(p_{\tau}^t,  p_{\tau})$, where $p_{\tau}$ and $p_{\tau}^t$ are the output probability of the student model and the pre-trained student model, reshaped by a temperature $\tau$. For our methods, we simply add 1-order perturbation to $p_{\tau}^t$, the $\epsilon$ is selected from $\{0.1, 0.2, 0.5\}$. The results are shown in the above table.
> >
> > To summarize, besides the previously added baselines (**Flooding** and **FilterKD**) and experiments (a **multi-class dataset** and a **synthetic Gaussian dataset**), we further compare our method with **Tf-KD** on **CIFAR-100** with several different models to make the evaluation more comprehensive.
> > For the benefit claim, we would like to make our statement more specific in the paper. We have always been humble to claim the benefits within the discussed settings and tasks. With the newly conducted experiments in the rebuttal stage, we found our method may also work for more diverse tasks. We will introduce these results in the next version while making our benefit claim clear.
> >
> > We appreciate it if the reviewer can recognize our efforts in the response, while also paying attention to our methodology and theoretical parts, where we present the systematical perturbation strategy and theoretical analysis regarding the principle of PTLoss and its connection to other methods.  Thanks in advance!

---

### Official Review · Reviewer_yv5C · 2022-10-25

**Confidence:** 2
**Clarity, Quality, Novelty And Reproducibility:** This paper is good enough in all four…
**Correctness:** 4
**Technical Novelty And Significance:** 4
**Empirical Novelty And Significance:** 4
**Recommendation:** 6

**Strength And Weaknesses:**

The proposed algorithm is systematic, well-organized, and has a solid theoretical background. This paper goes beyond simply pertuㄷrbating the distribution of teachers and proposes the best perturbation strategy, and several results presented in the experimental results demonstrate that this strategy is effective.

Despite the advantages mentioned above, I believe the experiment was conducted only in a relatively narrow space. Empirical validation for more diverse scenarios will be more helpful in demonstrating the effectiveness of the algorithm.
First of all, how does this algorithm work with students of larger capacity? Some KD algorithms test not only if a student has a lower capacity than the teacher but also if they have the same or greater capacity.
Next, how does the proposed algorithm work if the teacher is not a single network but an ensemble of multiple networks?
Finally, I wonder how the proposed algorithm can work for images, which is another big area where KD is applied.


---
After reading the reviews of other PCs and the authors' replies,

I lower my primary score to 6.
I still think this paper is well written and has enough contributions to be accepted, but as reviewer BGUG points out, this paper does not sufficiently cover state-of-the-art KD algorithms. For example, it would have been nice to have compared more recent methods in Table 5.1, as well.
Also, the application of the proposed algorithm is limited at this point. As the authors also mentioned, it would be good to explore computer vision further, the effectiveness of the proposed PTLoss according to the relationship between student and teacher capacity, etc., in future work.

---

**Summary Of The Paper:**

In this paper, the authors propose a KD algorithm that can improve the generalizability of students through perturbation of the teacher's output distribution. The PTLoss proposed in this paper is obtained by expressing a KL-based loss function through a Maclaurin series and then perturbing the terms of the preceding order in this series. The paper shows theoretically and experimentally that this new loss function can show higher KD performance than other competitors.

**Summary Of The Review:**

Overall, I read this paper interesting and lean to positive. Additional experiments will help to confirm the effectiveness of the proposed algorithm.

---

> ### Author Response · Authors · 2022-11-19
> **Author Response to Reviewer yv5C**
>
> We appreciate the reviewer likes our idea.
>
> For the empirical validation for more diverse scenarios, we do incorporate some in our updated draft.
> Specifically, we
> 1. extend our method to multi-class problems and validate its effectiveness on a multi-class classification task;
> 2. construct a synthetic Gaussian dataset to study if our claim holds for the known ground truth distribution.
>
> These scenarios and results help us to demonstrate the effectiveness of PTLoss.
>
> Meanwhile, we believe the suggested settings are also worthy of exploring. For example, the KD from the ensemble of networks is actually our ongoing research. By adding perturbation to each member of the ensemble, we can manipulate the teacher models to promote some desired properties such as calibration and teacher diversity. We look forward to presenting the follow-up works in later venues.
>
> This algorithm can work with different-capacity students. In fact, we care about the capacity difference between the student model and the teacher model, more than the capacity change from a single side. Thus it should be equivalently for validating the effectiveness of our method by varying the teacher's capacity as we showed in the main results. We understand that some works study KD for teachers and students of the same size and it is an interesting setting. However, our method focuses more on the model compression scenario, which is more applicable in the recent decade - we are provided with huge pre-trained models and want to deploy a lightweight model on edge devices with limited computational resources.
>
> For the image-related tasks, we noticed that reviewers showed interest without exception. Due to the time limit, we humbly introduce our method only in text-related tasks, and we welcome other researchers with more experience in the CV field can explore the potential application of PTLoss.

---

> ### Author Response · Authors · 2022-12-13
> **Response to the follow-up comments from Reviewer yv5C**
>
> We appreciate the follow-up response from Reviewer yv5C.
> 1. For the SOTA KD algorithm, we have included **FilterKD** and **Flooding** in our baselines (**A.5** in the updated paper draft) per the reviewers' suggestions in the first-round response. Another suggested baseline **Tf-KD** uses a totally different setting compared to ours, but considering reviewer BGUG’s comments in the second-round, we now further add Tf-KD in the comparison by making a fair setting for both Tf-KD and our method.
>
> Besides the previously added experiments (a **multi-class task** and a **synthetic Gaussian dataset** in A.6 and A.7), we also explored the computer vision task **CIFAR-100** in this round. The results are shown in the following table. We found PTLoss can still outperform those baselines, even they are designed specifically for the vision tasks.
> | Model | Teacher Acc. | Baseline| Tf-KD	| PTLoss |
> | :--- | :----: | :----: | :----: | ----: |
> | ResNet18 | 76.03 | 75.87 | 76.65 | **77.48** |
> | GoogleNet | 78.31 | 78.72 | 79.64 | **80.22** |
> | DenseNet121 | 79.04 | 79.04 | 79.58 | **80.12** |
>
> Note: The baseline results are same as in Table 5 of the Tf-KD paper(Yuan et al., 2020). The Tf-KD results are implemented by us. It is different from the original paper because we disable the ground truth data during the distillation stage.
> The **Tf-KD** implementation is modified from **Tf-KD_self** to be incorporated into our setting. Specifically, we modified Eq.(7) in the Tf-KD paper (Yuan et al., 2020) as $L_{self} = D_{\text{KL}}(p_{\tau}^t,  p_{\tau})$, where $p_{\tau}$ and $p_{\tau}^t$ are the output probability of the student model and the pre-trained student model, reshaped by a temperature $\tau$. For our methods, we simply add 1-order perturbation to $p_{\tau}^t$, the $\epsilon$ is selected from $\{0.1, 0.2, 0.5\}$.
>
> 2. For the relationship between student and teacher capacity, we have shown different-capacity teachers or students in both our original draft and the added CIFAR-100 experiments. On the text-related datasets, we use **T5-xxl**, **T5-xl**, and **T5-large** for the teacher models. On the image-related dataset, we use **ResNet18**, **GoogleNet**, and **DenseNet121** for the teacher and student models. Besides the consistent performance improvement, we would like to further emphasize that such improvement comes from the correction to the teacher’s output distribution, regardless of the capacity. Our experiments in **A.7** in the updated paper draft demonstrate this point.
>
> 3. We hope reviewer yv5C may reconsider the recommendation score based on our second-round response to reviewer BGUG’s comments. We think the main concerns that remained in the first round have been covered in this round, i.e., the comparison with more SOTA methods, the experiments on vision tasks, as well as the results with different-capacity students or teachers. Also, we hope the reviewers can pay more attention to our contributions made on the method and theory parts, where we present the perturbed loss with a proxy teacher method to systematically search the perturbation coefficients, and the theoretical proof that PTLoss can reduce the deviation from the population risk. We will appreciate reviewer yv5C raising back the recommendation or any further suggestions.

---

### Author Response · Authors · 2022-11-19
**Summary of Revision**

We thank all the reviewers for their feedback and support. We have edited the paper to incorporate the helpful reviews, including additional experiments, illustrative context, refined notations, and more related works. Here we provide a general response to summarize the major revisions made in the updated paper draft:
1. Extended the method to multi-class settings in Sec. 4 and added corresponding experiments on a multi-class dataset in A.6.
2. Added two baselines mentioned by reviewers, Flooding (Ishida et al., 2020) and FilterKD (Ren et al., 2022), shown in A.5.
3. Reported multiple-run results in A.5.
4. Added an experiment on a synthetic Gaussian dataset in A.7.
5. Added more illustrative text for some equations mentioned by reviewers.
6. Elaborated more on the proxy teacher part (Sec. 4.1.2 - Sec. 4.1.3)
7. Cited more related works.

We hope the updated content can more clearly illustrate our methods and more strongly support our conclusions, and thus address reviewers' concerns. We appreciate you taking to look through our updated draft and response.  We hope you may kindly consider raising your score in light of our discussion.

---

> ### Author Response · Authors · 2022-11-19
> **Supplementary Draft**
>
> Dear PCs and reviewers,
>
> We are aware of **the updated draft is not shown in the OpenReview system**, though we are certain we made the submission before the stage 1 deadline. **We thus uploaded the draft to this anonymous link**. You can find the updated draft via: **https://drive.google.com/file/d/1uxSDMNl9CfXatCc_zeXVgk7pZQMBYuJK/view?usp=sharing.**
>
> **This updated draft includes important revisions** as mentioned above, we hope it can help convey more detailed information to the reviewers. Also, we have contacted the PCs to solve the submission issues and will keep you updated regarding the progress.

---

> ### Author Response · Authors · 2022-12-13
> **Additional Revision after the Follow-up Response**
>
> We appreciate the reviewers' engagement after our first-round response. Per the reviewer's suggestions, we further add one CV dataset CIFAR-100, and a baseline Tf-KD. The results are shown in the following table.
>
> | Model | Teacher Acc. | Baseline | Tf-KD	| PTLoss |
> | :--- | :----: | :----: | :----: | ----: |
> | ResNet18 | 76.03 | 75.87 | 76.65 | **77.48** |
> | GoogleNet | 78.31 | 78.72 | 79.64 | **80.22** |
> | DenseNet121 | 79.04 | 79.04 | 79.58 | **80.12** |
>
> Totally, we added 3 experiments on the **CIFAR-100**, *the *multi-class MNLI**, and a **synthetic Gaussian dataset**, and included 3 more baselines **Tf-KD** (Yuan et al., 2020), **Flooding** (Ishida et al., 2020), and **FilterKD** (Ren et al., 2022) in the rebuttal stage. The other updates can be found in the **summary of revision** above.
>
> Besides, we hope the committee may also pay attention to our contributions made in the methodology and theory parts. We present a well-motivated perturbed loss to improve the KL loss in KD, and design a systematic strategy to search the perturbation coefficients. Theoretically, we illustrate how the perturbed loss can reduce the deviation from the true population risk and present the connection between PTLoss and label smoothing. We empirically show the effectiveness of our method from both real-world and synthetic datasets, spanning across text-related tasks and image-related tasks.
>
> Again, we thank all the reviewers' and ACs' time and feedback, which helps us further polish the paper and improve its impact.

---

### Decision · Program_Chairs · 2023-01-20

**Decision:**

Reject

**Justification For Why Not Higher Score:**

The concerns on the evaluation part are valid, i.e., recent strong KD baselines and diverse tasks are needed. In particular, I think including latest KD baselines is even more important than conducting  experiments on diverse tasks (e.g., image tasks). Overall, I recommend rejection for this paper.

**Justification For Why Not Lower Score:**

N/A

**Metareview: Summary, Strengths And Weaknesses:**

In this paper, the authors proposed to perturb the KL divergence between the teacher and student outputs. Specifically, they proposed PTloss by perturbing the leading terms in the Maclaurin series of KL divergence. Experiments on multiple NLP datasets show the effectiveness of PTloss.

Strengths:
1. PTloss is new. The authors also show how this perturbation subsumes label smoothing as a special case.
2. Both empirical and theoretical analyses are provided. The results on NLP datasets are good.

Weaknesses:
1. The authors should conduct experiments on diverse datasets to support the benefit claim. Otherwise, the authors need to narrow down the application scenarios in Abstract and Related work.
2. More KD baselines should be included. For example, Tf-KD is a special LS. In addition to vanilla LS, the authors should also compare PTLoss with a more advanced LS - Tf-KD.


**Summary Of Ac-Reviewer Meeting:**

We had a virtual meeting on 9 Dec with all the three PC members.

Reviewer Djov explained that the authors had address many of his concerns and that was why he had already increased the rating to 6. He also mentioned that ideally the authors should include more baselines and more datasets. However, given the novelty of the method, he is ok with current evaluation.

Reviewer BGUG still had the concerns on the evaluation part, i.e., recent strong KD baselines and diverse tasks are needed. Even if the authors focus on NLP tasks, larger datasets like SuperGLUE should be used.

Reviewer yv5C also agreed more evaluations are needed and thus he will decrease his rating.